# Predicting Pro-Environmental Behavior: The Leading Influence of Environmental Attitudes

**DOI:** 10.3390/bs15030291

**Published:** 2025-03-02

**Authors:** Velina Hristova, Kaloyan Haralampiev, Ivo Vlaev, Sonya Karabeliova

**Affiliations:** 1Institute for Population and Human Studies, Bulgarian Academy of Sciences, 1113 Sofia, Bulgaria; 2Psychology Department, Sofia University “St. Kliment Ohridski”, 1504 Sofia, Bulgaria; 3Department of Sociology, Faculty of Philosophy, Sofia University “St. Kliment Ohridski”, 1504 Sofia, Bulgaria; k_haralampiev@phls.uni-sofia.bg; 4Warwick Business School, University of Warwick, Coventry CV4 7AL, UK; ivo.vlaev@wbs.ac.uk; 5Department of Psychology, Faculty of Philosophy, Sofia University “St. Kliment Ohridski”, 1504 Sofia, Bulgaria

**Keywords:** pro-environmental behavior, environmental attitudes, personality traits, self-efficacy

## Abstract

This study explored the psychological determinants of pro-environmental behaviors through a multidimensional approach, integrating the roles of self-efficacy, personality traits and environmental attitudes (verbal, actual and affective commitment). While previous research has primarily examined general ecological engagement, this study differentiated between three specific domains: general ecological behaviors, prosocial and volunteering actions in the ecological context, and household chemical usage. A sample of 669 participants completed measures assessing the Big Five personality traits, self-efficacy and environmental commitment dimensions. Regression analyses revealed that environmental attitudes were the strongest predictors of general ecological behavior, while actual commitment, self-efficacy and extraversion played key roles in prosocial engagement and volunteering in the ecological context. Neuroticism and extraversion emerged as the strongest predictors of household chemical usage. The study suggests that the influence of personality traits on pro-environmental behavior may be context-dependent, with certain traits playing a more significant role when the behavior involves direct social engagement. Additionally, our findings emphasize the importance of fostering environmental commitment as a key driver of sustained ecological behavior, rather than relying solely on personality-driven tendencies or self-efficacy.

## 1. Introduction

### 1.1. Environmental Attitudes

Maloney et al. conceptualized environmental attitudes as comprising three key components: verbal commitment, actual commitment and affective commitment ([21]; [22]). Verbal commitment refers to what individuals state they are willing to do to address ecological concerns, such as supporting pro-environmental policies or pledging to adopt sustainable behaviors. However, stated intentions do not always translate into concrete action ([21]). Actual commitment, on the other hand, measures an individual’s real-world behaviors, such as recycling, using public transportation, or reducing consumption of polluting products. Studies have shown that actual commitment is often lower than verbal commitment, indicating a gap between what people say they will do and what they actually practice ([22]). Affective commitment assesses the emotional investment in environmental issues, such as frustration over pollution or distress about ecological degradation.

While affective commitment has long been recognized in psychology as a key dimension of environmental attitudes ([21]; [22]), in recent years, eco-anxiety has emerged as the dominant term to describe the intense emotional reactions specifically linked to climate change and environmental degradation. While affective commitment, as used by Maloney, encompasses a general emotional concern for ecological issues, eco-anxiety highlights the psychological distress, fear and uncertainty associated with the accelerating climate crisis ([1]; [7]). This evolution in terminology reflects not only an increased awareness of environmental threats but also a recognition of their mental health implications, reinforcing the strong connection between emotional engagement and ecological behavior.

A study by Fraj and Martinez examined ecological consumer behavior by analyzing the relationship between environmental attitudes and pro-environmental actions ([8]). Using the three-dimensional approach—which includes affective commitment, verbal commitment and actual commitment—the study investigated how emotional attachment, stated intentions and real behaviors contribute to sustainable consumer choices. The findings suggest that emotional concern for environmental issues has a stronger influence on behavior than verbal commitment. However, there is a clear gap between verbal and actual commitment, meaning that while many individuals express pro-environmental attitudes, fewer follow through with consistent sustainable behaviors.

### 1.2. Self-Efficacy

Self-efficacy, a key concept in social cognitive theory, is defined by Albert Bandura as “the belief in one’s capabilities to organize and execute the courses of action required to produce given attainments” ([4]). In the context of pro-ecological behavior, self-efficacy refers to an individual’s confidence in their ability to take effective environmental actions, such as reducing waste, conserving energy, or advocating for sustainability. Research has consistently shown that higher self-efficacy enhances engagement in sustainable behaviors, though its influence varies depending on context and external factors.

Several studies have explored the direct and indirect effects of self-efficacy on environmental engagement. Tagkaloglou and Kasser examined self-efficacy in activist pro-environmental goal-setting, finding that self-determined motivation and self-efficacy strongly predicted goal progress among students participating in collaborative environmental activism ([34]). The study also highlighted that Motivational Interviewing was more effective for individuals already engaged in environmental behaviors, whereas a more directive approach worked better for those less ready to change, underscoring the importance of tailored interventions to enhance self-efficacy. Similarly, Putrawan and Ningtyas found a significant correlation between self-efficacy and pro-environmental behavior in high school students in Jakarta, suggesting that fostering a sense of competence and confidence is crucial in shaping sustainable habits in younger populations ([27]).

Beyond direct behavioral outcomes, self-efficacy has also been identified as a critical mediator in the relationship between climate anxiety and action. While climate change anxiety can sometimes drive pro-environmental behaviors, it can also result in eco-paralysis, a state of inaction due to overwhelming fear ([14]). This study found that higher self-efficacy mitigates eco-paralysis, enabling individuals to channel their anxiety into proactive behaviors rather than avoidance. Such findings suggest that building self-efficacy could be a key strategy in transforming environmental concern into constructive engagement.

The role of self-efficacy in environmental behavior spillover has also been demonstrated. Lauren et al. found that engaging in simple, low-effort pro-environmental behaviors, such as water conservation, increased self-efficacy, leading individuals to adopt more challenging behaviors, such as installing water-efficient appliances ([20]). This supports the idea that gradually increasing self-efficacy through small, achievable actions can foster long-term pro-ecological commitment. Similarly, Huang explored the impact of media exposure and found that self-efficacy mediates the relationship between environmental beliefs and behavior ([13]). The study suggests that individuals who are exposed to global warming media coverage and feel confident in their ability to act are more likely to engage in pro-environmental behaviors, emphasizing the importance of strategic communication and education in strengthening self-efficacy.

However, research on the relative importance of self-efficacy compared to environmental attitudes presents mixed findings. Yoong et al. found that higher self-efficacy strengthens pro-environmental attitudes and encourages sustainable behaviors, supporting the idea that enhancing self-efficacy through intervention programs can drive ecological engagement ([35]). In contrast, a large-scale international study across 11 countries (N = 11,000) by Miller et al. revealed that environmental attitudes were a stronger predictor of pro-environmental behavior than self-efficacy, which had only a minor direct effect and no significant moderation role ([25]). These findings suggest that while self-efficacy contributes to pro-ecological behavior, its impact may be secondary to environmental attitudes, highlighting the need for context-specific strategies to maximize its influence.

### 1.3. Personality Traits

With the growing environmental challenges, environmental psychologists and sociologists have started to study the pro-environmental individual—a person consistently engaged in sustainable actions across domains like energy use, water consumption and waste reduction—by examining characteristics linked to pro-environmental behavior including personality traits ([23]).

Some of the earliest studies investigating the relationship between personality traits and pro-environmental behavior found a significant positive direct relationship between pro-ecological actions and the traits of agreeableness and openness to experience ([12]; [26]; [11]). Further research confirmed the indirect relationship with these traits mediated through the sense of connectedness with nature ([23]). The role of agreeableness was confirmed further in several studies ([24]; [19]). However, a study examining personality traits and behaviors related to a specific type of environmental behavior, reducing greenhouse gas emissions, did not confirm these findings ([6]). The authors did not find an effect for agreeableness, despite using the extended HEXACO personality model ([3]), in which agreeableness is distinguished from the Honesty-Humility subscale. In their study, openness to experience and conscientiousness had the strongest effects, with their influence being mediated by pro-environmental attitudes.

Recent studies confirm that openness, agreeableness and conscientiousness show the strongest correlations with pro-environmental behavior, while neuroticism is generally not significant ([31]). Finally, a meta-analysis of 38 studies further highlights openness and Honesty-Humility as the strongest predictors, while agreeableness, conscientiousness and extraversion show weaker effects, and neuroticism remains non-significant ([32]).

Given the escalating environmental crisis and the need for behavioral changes at both individual and societal levels, understanding the psychological determinants of pro-environmental behavior is crucial. The aim of the current study is to explore pro-environmental behavior through a multidimensional approach, examining how personality traits, self-efficacy and environmental attitudes (affective, actual and verbal commitment) interact to shape various types of pro-ecological behaviors. Unlike previous research that has primarily examined general environmental engagement, this study differentiates between three distinct categories of pro-environmental behaviors: general behaviors, pro-social and volunteering actions, and household chemical usage. In doing so, it expands upon existing literature by considering how personality-driven tendencies interact with self-efficacy and environmental attitudes to predict various pro-environmental actions. By integrating personality, self-efficacy and multiple dimensions of environmental commitment, this study aims to provide a comprehensive framework for understanding the psychological mechanisms underlying pro-ecological behavior. The findings are expected to contribute to behavioral interventions, sustainability education and policy development, helping to design more effective, personality-informed strategies for promoting environmental responsibility.

## 2. Materials and Method

### 2.1. Participants and Procedure

The study was conducted online with a total of 669 participants in a cross-sectional design. The sample consisted predominantly of women (60.5%), with participants ranging in age from 15 to 77 years (M = 29.30, SD = 13.39). Recruitment was carried out through Sofia University’s outreach campaign, utilizing social media, posters, and other promotional methods to engage potential respondents. Participation was anonymous and voluntary, with no financial compensation provided. Ethical approval was obtained (No. 38/21 June 2024) in compliance with Sofia University’s Research Code of Practice.

### 2.2. Materials/Instruments

#### 2.2.1. Big Five Personality

The Big Five personality traits were measured using a shortened version of the P. John and S. Srivastava questionnaire ([15]), adapted for Bulgarian-speaking participants ([33]). This version included three items per trait, totaling 15 statements across five dimensions: extraversion, agreeableness, conscientiousness, neuroticism, and openness to experience. Participants rated each item on a five-point Likert scale, ranging from “Strongly Disagree” (1) to “Strongly Agree” (5).

#### 2.2.2. Self-Efficacy

The Self-Efficacy Scale was developed by Sherer et al. and consists of ten statements that assess an individual’s perception of their own capabilities ([29]). Participants rate each statement on a four-point Likert scale, ranging from “Absolutely False” (1) to “Absolutely True” (4).

#### 2.2.3. Environmental Attitudes

To assess environmental attitudes, this study employed the Environmental Consumer Behavior Survey, a questionnaire originally based on the Maloney et al. framework and later adapted into a more contemporary version by Fraj and Martinez ([22]; [8]). The Bulgarian adaptation of the survey consists of 23 self-assessment items rated on a five-point Likert scale (1 = Strongly Disagree to 5 = Strongly Agree). The questionnaire identifies three distinct factors: affective commitment, verbal commitment, and actual commitment. Verbal commitment measures individuals’ stated intentions toward environmental engagement, while actual commitment captures their real pro-environmental behaviors. Affective commitment, on the other hand, reflects their emotional responses and concerns regarding ecological issues.

#### 2.2.4. General Measure of Ecological Behavior

The questionnaire was developed by Florian Kaiser ([16]; [18]) and measures various aspects of ecological behavior. It consists of 37 self-assessment statements using a five-point Likert scale ranging from “Never-1” to “Very Often-5”. In its original version, seven factors are identified: prosocial behavior, ecological garbage removal, water and power conservation, ecologically aware consumer behavior, garbage inhibition, volunteering in nature protection activities, and ecological automobile use.

While the original version of the questionnaire comprised seven factors, the current study identified six factors through Principal Component analysis. Three of these scales demonstrated acceptable validity (see Table A1 in Appendix A), while the remaining three had Cronbach’s Alpha values below 0.4 and were therefore excluded. Additionally, items with factor loadings below 0.4 were removed from the retained scales. The final version of the questionnaire, containing only statistically valid items, is presented in Table A2 in Appendix A.

The first scale, labeled “General Ecological Behaviors,” encompasses a broad range of pro-environmental actions. The second scale focuses on pro-social behaviors and environmental volunteering, while the third scale assesses household chemical usage. Unlike the first two scales, which represent a unique combination of statements derived from the current sample, the third scale remains identical to the corresponding scale in the original questionnaire.

Several international studies have validated the General Ecological Behavior (GEB) scale across diverse cultural contexts, demonstrating both its reliability and the influence of situational factors on ecological engagement. Research comparing Swiss and Swedish samples confirmed the scale’s applicability but highlighted how structural differences between countries shape ecological behaviors, with certain actions being easier to adopt in one society than the other ([17]). Similarly, studies in Germany, Argentina, and the United States found that while the GEB scale effectively measures ecological behavior across cultures, the extent of engagement varies due to societal norms, infrastructure, and policy environments ([18]; [10]). Additionally, research in Germany using actual electricity consumption data demonstrated the criterion validity of the GEB scale, linking self-reported behavior to real-world ecological impact ([2]). These findings underscore the scale’s cross-cultural adaptability while emphasizing that local infrastructure, policies, and socio-economic conditions significantly influence the extent to which individuals can engage in ecological behaviors. In line with this research, our study confirmed the applicability of the GEB scale in a Bulgarian sample, while also identifying modifications in factor structure that reflect the influence of local infrastructure, policies, and socio-economic conditions on ecological behavior.

## 3. Results

The scales used in the study demonstrated internal consistency, with Cronbach’s α ranging from 0.60 to 0.80 (See Table 1).

Following the normality analysis of the distribution, correlation analyses were performed using Pearson’s coefficient to evaluate the relationships. Significant correlations were identified and are shown in Table 2.

### 3.1. General Ecological Behavior

A regression analysis investigated the predictors of general ecological behavior, incorporating personality traits, environmental attitudes, self-efficacy, and commitment dimensions (Table 3). The model accounts for 34.8% of the variance (R^2^ = 0.348, Adjusted R^2^ = 0.339), indicating moderate explanatory power. The ANOVA results (F = 39.122, *p* < 0.001) confirm that the overall model is statistically significant, suggesting that at least some of the included predictors have meaningful contributions to ecological behavior.

Among the predictors, verbal commitment (β = 0.288, *p* < 0.001, power = 1.000) emerged as the strongest factor, implying that individuals who express strong verbal commitment are significantly more likely to engage in ecological behaviors. Similarly, actual commitment (β = 0.274, *p* < 0.001, power = 1.000) also plays a crucial role, reinforcing the idea that real-life sustainable actions are strong indicators of broader ecological engagement. Affective commitment (β = 0.102, *p* = 0.013, power = 0.802) was also found to be a significant predictor, suggesting that emotional engagement with environmental issues contributes to pro-environmental behavior, although its power is slightly lower than verbal and actual commitment.

In contrast, personality traits demonstrated weaker and more inconsistent effects. Extraversion (β = −0.087, *p* = 0.015, power = 0.793) showed a small but statistically significant negative effect, indicating that more extroverted individuals may be less engaged in general ecological behavior. However, its low power suggests caution in interpreting this finding. Other personality traits, including conscientiousness, agreeableness, neuroticism, and openness, were non-significant predictors with low power, suggesting that they do not have a direct impact on general ecological behavior within this model. Self-efficacy was also non-significant, with a very low power value, indicating that an individual’s confidence in their ability to act does not directly predict ecological behavior.

### 3.2. Prosocial Behavior and Volunteering in the Ecological Context

A regression analysis examined the predictors of prosocial behavior and volunteering in the ecological context, incorporating personality traits, environmental attitudes, self-efficacy, and commitment dimensions (Table 4). The model explains 33.8% of the variance (R^2^ = 0.338, Adjusted R^2^ = 0.328), indicating moderate explanatory power. The ANOVA results (F = 37.303, *p* < 0.001) confirm that the overall model is statistically significant.

Among the predictors, actual commitment (β = 0.470, *p* < 0.001, power = 1.000) emerged as the strongest and most reliable predictor, indicating that individuals who already engage in real-world ecological behaviors are significantly more likely to participate in prosocial and volunteering activities. In addition to actual commitment, self-efficacy (β = 0.121, *p* = 0.002, power = 0.939) and extraversion (β = 0.091, *p* = 0.012, power = 0.814) were also significant predictors with sufficient statistical power. Other predictors, including neuroticism (β = 0.073, *p* = 0.027, power = 0.720) and verbal commitment (β = 0.089, *p* = 0.048, power = 0.636), showed marginal significance but lacked sufficient statistical power, suggesting that their effects may be less reliable.

### 3.3. Household Chemical Usage

A regression analysis examined the predictors of household chemical usage, incorporating personality traits, environmental attitudes, self-efficacy, and commitment dimensions (Table 5). The model explains 11.1% of the variance (R^2^ = 0.111, Adjusted R^2^ = 0.099), indicating low explanatory power. The ANOVA results (F = 9.186, *p* < 0.001) confirm that the overall model is statistically significant. However, given the low R^2^ value, a substantial proportion of the variance remains unexplained, suggesting that other factors not included in this model may play a larger role.

Among the predictors, neuroticism (β = 0.213, *p* < 0.001, power = 1.000) and extraversion (β = 0.185, *p* < 0.001, power = 0.998) were the strongest and most reliable predictors. Other predictors, including self-efficacy (β = 0.108, *p* = 0.014, power = 0.797), showed marginal significance but lacked sufficient power, suggesting that confidence in one’s ability to act does not consistently predict household chemical usage. Verbal commitment (β = −0.121, *p* = 0.020, power = 0.758) and actual commitment (β = −0.099, *p* = 0.037, power = 0.678) were both negatively associated with household chemical usage, indicating that individuals who report higher environmental commitment may consciously avoid or reduce their use of household chemicals. However, their power values are below 0.8, making these findings less reliable.

## 4. Discussion

The results of the empirical study revealed that environmental attitudes (covering verbal, actual, and affective commitment) are the most significant predictors of general ecological behavior. Individuals who express a strong intention to engage in sustainability practices, who already take pro-environmental actions or are emotionally engaged with ecological topics are the most likely to adopt and maintain environmentally friendly behaviors. These findings align with previous research, which also identified environmental attitudes (defined in their study as valuing nature and environmental concerns) as a stronger predictor of pro-environmental behavior than self-efficacy ([25]). In contrast, personality traits did not appear significant and with enough statistical power to predict general pro-ecological behavior in our sample. Our results do not support prior research that found agreeableness, openness and conscientiousness to be significant predictors of pro-environmental behavior ([12]; [26]; [11]; [24]; [19]). This divergence suggests that while personality traits may play a role in shaping environmental engagement, their influence may be more indirect or context-dependent than previously assumed.

When examining prosocial behaviors and volunteering in the ecological context, our study found that actual commitment to ecological behaviors, extraversion, and self-efficacy were the strongest predictors. This supports the spillover effect hypothesis, where engaging in one form of environmental behavior fosters broader prosocial actions ([6]). However, our findings do not fully align with prior studies on personality and pro-environmental behavior. While a meta-analysis of 38 studies found weak or inconsistent effects for extraversion ([31]), our study identified extraversion as a significant predictor within the specific context of prosocial and volunteering behavior. This may be due to the inherent social nature of volunteering, where communication, interaction, and group engagement play a central role, making extraverted individuals more likely to participate. These findings further suggest that the role of personality traits in pro-environmental behavior may be context-dependent, with certain traits becoming more relevant when the behavior involves direct social engagement.

Our results also provide insights into the role of self-efficacy in pro-environmental behavior, aligning with existing literature while highlighting some discrepancies. Studies have consistently shown that higher self-efficacy enhances pro-ecological engagement ([34]) and that self-efficacy can mitigate eco-paralysis caused by climate anxiety ([14]). Our findings support the idea that self-efficacy plays a significant role in prosocial behaviors and volunteering in the ecological context, reinforcing previous research suggesting that individuals who feel confident in their ability to contribute to environmental change are more likely to engage in sustainability efforts ([27]). However, our results indicated that self-efficacy did not significantly predict general ecological behavior, which aligns with large-scale studies showing that environmental attitudes are often stronger predictors than self-efficacy ([25]). This suggests that while self-efficacy is crucial in some contexts, particularly in prosocial behavior and volunteering, its impact may be secondary to commitment-based factors in predicting overall ecological engagement.

For household chemical usage, neuroticism and extraversion stand out as the most reliable predictors. This finding may be explained by the positive correlation between neuroticism and obsessive compulsive behaviors, including heightened concerns about cleanliness and frequent disinfecting practices, which have been linked to anxiety-driven tendencies ([9]; [5]; [28]). Individuals high in neuroticism may perceive household cleanliness as a way to mitigate stress or environmental threats, leading to increased use of household chemicals. Extraversion is also positively associated with chemical use, suggesting that more socially active individuals may use these products more frequently, perhaps due to lifestyle factors requiring more frequent cleaning or self-presentation. Studies on behavior during the COVID-19 pandemic further support this relationship, indicating that highly extraverted individuals were more likely to engage in rigorous cleaning and disinfecting routines as part of their social interactions and risk management strategies ([30]). These findings suggest that both personality-driven anxiety and social engagement may contribute to increased household chemical consumption, highlighting the complex interplay between psychological traits and environmental behaviors.

## 5. Conclusions, Implications and Future Directions

Our study showed that environmental attitudes are the strongest predictors of general ecological behavior, whereas actual commitment, self-efficacy and extraversion are the strongest predictors of prosocial engagement and volunteering in the ecological context. However, our results challenge previous claims about the role of personality traits, suggesting that environmental attitudes may be stronger predictors of sustainability-related behaviors than personality alone. Additionally, while self-efficacy is a key factor in prosocial environmental actions, it does not significantly predict general ecological behavior, supporting the idea that environmental attitudes and direct commitment are often more influential. Our findings suggest that the role of personality traits in pro-environmental behavior may be context-dependent, with certain traits becoming more relevant when the behavior involves direct social engagement. Finally, our results emphasize the importance of fostering commitment-based interventions rather than relying solely on personality-driven tendencies or self-efficacy to promote sustainable behavior.

This study has several limitations that should be considered. First, the use of a brief assessment tool for measuring personality traits may have restricted the depth of analysis, as it does not capture the full complexity of trait subdimensions that could influence pro-environmental behaviors in distinct ways. Second, reliance on self-reported measures introduces the possibility of response biases, particularly social desirability bias, where participants may overstate their environmental behaviors or commitment. Additionally, the cross-sectional nature of the study limits causal inferences, making it difficult to determine whether self-efficacy and commitment drive pro-environmental behavior or if engaging in sustainability actions strengthens these psychological predictors over time. Longitudinal or intervention-based research would help clarify these relationships and assess changes in ecological behaviors over time. Finally, the study’s generalizability may be constrained by sample characteristics, as voluntary participation could have led to a self-selection bias favoring individuals already invested in environmental issues. Expanding research to include more diverse populations across different sociocultural and economic contexts would enhance the applicability of findings and provide a more comprehensive understanding of the psychological drivers of sustainability behaviors.

## Figures and Tables

**Table 1 behavsci-15-00291-t001:** Descriptive statistics and reliabilities.

Scales	Number of Items	M	SD	α
Extraversion	3	10.77	2.63	0.71
Agreeableness	3	11.62	2.27	0.62
Consciousness	3	11.95	2.42	0.74
Neuroticism	3	9.62	2.76	0.6
Openness to experience	3	11.46	2.41	0.75
Self-efficacy	10	29.8	5.05	0.87
Affective Commitment	7	26.28	5.36	0.8
Verbal Commitment	7	23.69	5.33	0.72
Actual Commitment	9	26.95	6.9	0.75
General ecological behaviors	7	19.45	5.07	0.65
Pro-social and volunteering behaviors	6	11.13	2.91	0.71
Ecologically aware household chemicals usage	6	16.29	3.82	0.67

**Table 2 behavsci-15-00291-t002:** Correlation analysis results.

	1	2	3	4	5	6	7	8	9	10	11	12
1. Affective commitment	1											
2. Verbal commitment	0.604 **	1										
3. Actual commitment	0.485 **	0.598 **	1									
4. Extraversion	0.156 **	0.094 *	0.134 **	1								
5. Agreeableness	0.168 **	0.206 **	0.157 **	0.276 **	1							
6. Conscientiousness	0.143 **	0.067	0.157 **	0.291 **	0.265 **	1						
7. Neuroticism	0.108 **	0.023	0.047	−0.144 **	−0.073	−0.114 **	1					
8. Openness	0.109 **	0.057	0.147 **	0.324 **	0.122 **	0.260 **	−0.012	1				
9. Self-efficacy	0.063	0.026	0.126 **	0.334 **	0.128 **	0.391 **	−0.205 **	0.414 **	1			
10. General ecological behavior	0.416 **	0.516 **	0.508 **	0.036	0.130 **	0.144 **	0.056	0.118 **	0.090 *	1		
11. Prosocial and volunteering	0.307 **	0.387 **	0.551 **	0.189 **	0.164 **	0.108 **	0.061	0.140 **	0.182 **	0.344 **	1	
12. Household chemical usage	0.04	−0.088 *	−0.069	0.195 **	0.091 *	0.110 **	0.158 **	0.065	0.123 **	−0.018	0.049	1

** Correlation is significant at the 0.01 level (2-tailed). * Correlation is significant at the 0.05 level (2-tailed).

**Table 3 behavsci-15-00291-t003:** Regression analysis results for general ecological behavior.

Predictor	B	SE	β	*t*	*p*	Lower CI	Upper CI	Power
Self-Efficacy	0.02	0.03	0.03	0.74	0.46	−0.04	0.08	0.18
Extraversion	−0.14	0.06	−0.09	−2.45	0.02	−0.24	−0.03	0.79
Agreeableness	0.02	0.06	0.01	0.26	0.80	−0.10	0.14	0.08
Conscientiousness	0.12	0.06	0.07	1.95	0.05	0.00	0.24	0.63
Neuroticism	0.04	0.05	0.03	0.84	0.40	−0.05	0.14	0.21
Openness	0.08	0.06	0.05	1.35	0.18	−0.04	0.20	0.39
Affective Commitment	0.08	0.03	0.10	2.48	0.01	0.02	0.14	0.80
Verbal Commitment	0.22	0.03	0.29	6.48	<0.001	0.15	0.29	1.00
Actual Commitment	0.16	0.02	0.27	6.75	<0.001	0.12	0.21	1.00

Note: B = unstandardized regression coefficient; SE = standard error; β = standardized coefficient; *t* = *t*-value; *p* = significance level; Lower CI and Upper CI represent 95% confidence intervals. Only predictors with a power value above 0.8 are considered reliable.

**Table 4 behavsci-15-00291-t004:** Regression analysis results for prosocial behavior and volunteering in the ecological context.

Predictor	B	SE	β	*t*	*p*	Lower CI	Upper CI	Power
Self-Efficacy	0.07	0.02	0.12	3.17	0.00	0.03	0.11	0.94
Extraversion	0.10	0.04	0.09	2.52	0.01	0.02	0.18	0.81
Agreeableness	0.07	0.04	0.05	1.50	0.13	−0.02	0.15	0.45
Conscientiousness	−0.06	0.04	−0.05	−1.33	0.18	−0.14	0.03	0.38
Neuroticism	0.08	0.04	0.07	2.21	0.03	0.01	0.15	0.72
Openness	−0.01	0.04	−0.01	−0.17	0.87	−0.09	0.08	0.07
Affective Commitment	0.00	0.02	−0.01	−0.13	0.90	−0.05	0.04	0.06
Verbal Commitment	0.05	0.03	0.09	1.98	0.05	0.00	0.10	0.64
Actual Commitment	0.20	0.02	0.47	11.47	<0.001	0.17	0.23	1.00

Note: B = unstandardized regression coefficient; SE = standard error; β = standardized coefficient; *t* = *t*-value; *p* = significance level; Lower CI and Upper CI represent 95% confidence intervals. Only predictors with a power value above 0.8 are considered reliable.

**Table 5 behavsci-15-00291-t005:** Regression analysis results for household chemical usage.

Predictor	B	SE	β	*t*	*p*	Lower CI	Upper CI	Power
Self-Efficacy	0.08	0.03	0.11	2.46	0.01	0.02	0.15	0.80
Extraversion	0.27	0.06	0.19	4.42	<0.001	0.15	0.39	1.00
Agreeableness	0.10	0.07	0.06	1.53	0.13	−0.03	0.23	0.46
Conscientiousness	0.07	0.07	0.04	1.05	0.30	−0.06	0.20	0.28
Neuroticism	0.30	0.05	0.21	5.56	<0.001	0.19	0.40	1.00
Openness	−0.07	0.07	−0.05	−1.07	0.29	−0.20	0.06	0.28
Affective Commitment	0.07	0.03	0.09	1.90	0.06	0.00	0.13	0.61
Verbal Commitment	−0.09	0.04	−0.12	−2.33	0.02	−0.16	−0.02	0.76
Actual Commitment	−0.06	0.03	−0.10	−2.09	0.04	−0.11	0.00	0.68

Note: B = unstandardized regression coefficient; SE = standard error; β = standardized coefficient; *t* = *t*-value; *p* = significance level; Lower CI and Upper CI represent 95% confidence intervals. Only predictors with a power value above 0.8 are considered reliable.

## Data Availability

Dataset available on request from the authors.

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
