# Peer review of "Predicting Pro-Environmental Behavior: The Leading Influence of Environmental Attitudes"

_behavsci, 2025, doi:10.3390/bs15030291_

Round 1
Reviewer 1 Report
Comments and Suggestions for Authors
This manuscript is well written, with detailed methodology and well defined references. Results are promising although the results of SEM can be better presented by a single flow-chart like diagram. The R^2 values of each constructs need to be presented and discussed.
This manuscript is unique because the respondents are from Bulgaria, a country not often seen in literature. Are the questions of the survey conducted in English or translated from another language into English? If any, does translation affect the understanding of the questions? It will be great if the authors can further discuss the different and similar of the Bulgaria results from other literature from Asia, Western Europe or America.
Author Response
Comment: This manuscript is well written, with detailed methodology and well defined references. Results are promising although the results of SEM can be better presented by a single flow-chart like diagram. The R^2 values of each constructs need to be presented and discussed.
Response: Thank you for your valuable feedback and for recognizing the strengths of our manuscript. Based on reviewers’ suggestions, we have carefully reviewed the entire methodology and reanalyzed the data to enhance the clarity and robustness of our findings. To improve the conceptual framework, we have now included all commitment dimensions (affective, verbal, and actual) and self-efficacy as part of the broader concept of the pro-environmental individual. As a result of these refinements, the analyses have been entirely reviewed. Instead of the original structural equation modeling (SEM), we now present several regression analyses that allow for a clearer interpretation of how personality traits, self-efficacy, and commitment dimensions contribute to different types of pro-environmental behaviors.
We also acknowledge your request for improved visualization of the results. While we have shifted away from SEM, we have made an effort to present the findings as clearly as possible, ensuring that all key relationships are systematically outlined. Additionally, the R² values for each regression model are stated.
Comment: This manuscript is unique because the respondents are from Bulgaria, a country not often seen in literature. Are the questions of the survey conducted in English or translated from another language into English? If any, does translation affect the understanding of the questions? It will be great if the authors can further discuss the different and similar of the Bulgaria results from other literature from Asia, Western Europe or America.
Response: Thank you for your thoughtful feedback. In response to your inquiry regarding the language of the survey and its potential impact on understanding, we clarify that the survey was conducted in Bulgarian, with the General Ecological Behavior (GEB) scale translated into Bulgarian. The translation process was carefully managed to ensure clarity and accuracy, following standard procedures for adapting psychological measures across languages. Specifically, items were translated and reviewed to maintain their conceptual meaning while ensuring linguistic appropriateness. Additionally, Principal Component Analysis (PCA) was conducted to validate the structure of the translated measure, resulting in a modified factor structure with three valid scales: General Ecological Behaviors, Prosocial and Volunteering Behaviors, and Household Chemical Usage. We added this information to the text.
Regarding comparisons with other international studies, previous research has demonstrated the cross-cultural applicability of the GEB scale, with validation studies conducted in Germany, Switzerland, Sweden, Argentina, and the United States. Based on your suggestion, we added more information in the text as follows:
“Several international studies have validated the General Ecological Behavior (GEB) scale across diverse cultural contexts, demonstrating both its reliability and the influence of situational factors on ecological engagement. Research comparing Swiss and Swedish samples confirmed the scale's applicability but highlighted how structural differences between countries shape ecological behaviors, with certain actions being easier to adopt in one society than the other [28]. Similarly, studies in Germany, Argentina, and the United States found that while the GEB scale effectively measures ecological behavior across cultures, the extent of engagement varies due to societal norms, infrastructure, and policy environments [29, 30]. Additionally, research in Germany using actual electricity consumption data demonstrated the criterion validity of the GEB scale, linking self-reported behavior to real-world ecological impact [31]. These findings underscore the scale's cross-cultural adaptability while emphasizing that local infrastructure, policies, and socio-economic conditions significantly influence the extent to which individuals can engage in ecological behaviors. In line with this research, our study confirmed the applicability of the GEB scale in a Bulgarian sample, while also identifying modifications in factor structure that reflect the influence of local infrastructure, policies, and socio-economic conditions on ecological behavior.”
Reviewer 2 Report
Comments and Suggestions for Authors
General review :
Rework the organization of the paper
Please could you bring a more detailled defintion of emotional engagement
In abstract
The abstract needs to be revised.
« This study investigates pro-environmental behavior through a personality-focused ap-10 proach as well as the role of emotional engagement with environmental issues, specifically through 11 affective commitment ... » Please briefly argue why it is important to explore emotionnal engagement and specifically affective commitmement
In the abstract, you do not explicitly state that you will be studying personality; this becomes clear only at the end when you mention traits.
Introduction
Revise the introduction. It does not introduce the problem or the question of why studying personality in relation to eco-anxiety and its connection to environmental behaviors is important. Rework the introduction and outline the elements you will present in the body of the article.
Emotions and pro-environmental behaviors
« Emotions and moods have the power to influence our choices and behaviors—on one 41 hand, by avoiding unpleasant feelings, and on the other hand, by enhancing positive ones ». It is unclear. Emotion avoid unpleasant feeling, But what about when it comes to negative emotions? Rework the sentence ;
You present the effects of emotions on environmental attitudes/behaviors, please bring some example from pas research. Illustrate the arguments.
Personality and pro-environmental behaviors
Rework the organization of the paragraphs.
« Recently, personality-oriented theories have emerged as part of the various attempts 64 to explore pro-environmental behavior. These approaches consider individuals as intri-65 cate systems composed of multiple interacting components, including personality traits 66 [32]. Therefore, we can explore why affective commitment motivates some people to en-67 gage in pro-environmental actions while others remain unaffected, and why behavior 68 change is driven by normative goals for some but not for others ».
The connection between personality theories and, in the next sentence, affective commitment is unclear. The sentences follow one another without clarity. Revise the organization of arguments and the link between these concepts.
You show that past research results using the Big Five, but are there other studies? If so, why not briefly present them, and make sure to justify the use of the Big Five rather than other personality theories.
Materials and Method
Participants and procedure
You mention using G*Power to determine the required sample size. This is very useful, but it would be helpful to briefly explain why you chose a small effect size (f² = 0.02) ?
You mention that you planned to exclude some participants, what are the criterion of exclusion ?
While you mention data collection through a university campaign and the use of Google Forms, it would be interesting to add more context regarding the duration of data collection ?
Have you considered the representativeness of the sample? If so, please developp it; if not, consider addressing it in the discussion.
Résults
« Previous studies have found links between Agreeableness and empathy, altruism, higher levels of selflessness, and greater concern for others [36, 37], which may explain the connection between this personality trait and pro-environmental behaviors. » Could you argument how is this connection possible ?
Comments on the Quality of English Languageno comment
Author Response
Comment: Rework the organization of the paper
Please could you bring a more detailled defintion of emotional engagement
Response: Thank you for your constructive feedback. In response to your suggestions, we have reworked the organization of the paper. To enhance the methodological rigor and analytical precision, we have carefully reviewed the entire methodology and reanalyzed the data. One major refinement is the inclusion of all commitment dimensions (affective, verbal, and actual) as well as self-efficacy as а component of the pro-environmental individual framework. As a result, we have revised our analyses, replacing the original structural equation modeling (SEM) with several regression analyses, which offer a clearer and more interpretable presentation of the relationships between personality traits, self-efficacy, and commitment dimensions.
To improve the structure of the manuscript and based on your comments, we have also reorganized the introduction, clearly outlining the study’s objectives and the key elements presented in the body of the paper (with separate subtitles for each one). This included the expansion of the conceptual distinctions between commitment dimensions, ensuring that each dimension—actual, affective, and verbal commitment—is clearly defined and distinguished. We hope that this revision not only improves the conceptual depth of the study but also directly addresses your request for a more clearer definition of emotional engagement.
Comment: In abstract
The abstract needs to be revised.
« This study investigates pro-environmental behavior through a personality-focused ap-10 proach as well as the role of emotional engagement with environmental issues, specifically through 11 affective commitment ... » Please briefly argue why it is important to explore emotionnal engagement and specifically affective commitmement
In the abstract, you do not explicitly state that you will be studying personality; this becomes clear only at the end when you mention traits.
Response: Thank you for highlighting the abstract. In response to your comments and the extensive revisions made throughout the paper based on reviewer feedback, we have rewritten the abstract as follows:
“This study explored the psychological determinants of pro-environmental behaviors through a multidimension approach, integrating the roles of self-efficacy, personality traits and environmental attitudes (verbal, actual, and affective commitment). While previous research has primarily examined general ecological engagement, this study differentiated between three specific domains: general ecological behaviors, prosocial and volunteering actions in the ecological context, and household chemical usage. A sample of 669 participants completed measures assessing the Big Five personality traits, self-efficacy, and environmental commitment dimensions. Regression analyses revealed that environmental attitudes were the strongest predictors of general ecological behavior, while actual commitment, self-efficacy and Extraversion played key roles in prosocial engagement and volunteering in the ecological context. Neuroticism and Extraversion emerged as the strongest predictors of household chemical usage. The study suggests that the influence of personality traits on pro-environmental behavior may be context-dependent, with certain traits playing a more significant role when the behavior involves direct social engagement. Additionally, our findings emphasize the importance of fostering environmental commitment as a key driver of sustained ecological behavior, rather than relying solely on personality-driven tendencies or self-efficacy.”
Comment: Introduction
Revise the introduction. It does not introduce the problem or the question of why studying personality in relation to eco-anxiety and its connection to environmental behaviors is important. Rework the introduction and outline the elements you will present in the body of the article.
Response: Thank you for your valuable comment. We agree with your suggestion and have implemented the revision as outlined in our previous responses. A separate subsection is now dedicated to environmental attitudes, which includes all commitment dimensions (actual, verbal, and affective). Additionally, we have provided distinct subsections for personality traits and self-efficacy, all examined within the ecological context.
Comment: Emotions and pro-environmental behaviors
« Emotions and moods have the power to influence our choices and behaviors—on one 41 hand, by avoiding unpleasant feelings, and on the other hand, by enhancing positive ones ». It is unclear. Emotion avoid unpleasant feeling, But what about when it comes to negative emotions? Rework the sentence ;You present the effects of emotions on environmental attitudes/behaviors, please bring some example from pas research. Illustrate the arguments.
Response: Based on our comprehensive revision of the paper, particularly the introduction, this sentence has been omitted. Additionally, we have shifted our focus to environmental attitude as a whole, covering all commitment dimensions (actual, verbal, and affective) rather than addressing emotional engagement separately. As a result, we have integrated the existing literature to reflect the full construct of environmental attitude.
Comment: Personality and pro-environmental behaviors
Rework the organization of the paragraphs.
« Recently, personality-oriented theories have emerged as part of the various attempts 64 to explore pro-environmental behavior. These approaches consider individuals as intri-65 cate systems composed of multiple interacting components, including personality traits 66 [32]. Therefore, we can explore why affective commitment motivates some people to en-67 gage in pro-environmental actions while others remain unaffected, and why behavior 68 change is driven by normative goals for some but not for others ».
The connection between personality theories and, in the next sentence, affective commitment is unclear. The sentences follow one another without clarity. Revise the organization of arguments and the link between these concepts.
Response: Based on our comprehensive revision of the paper, this paragraph has been omitted.
Comment: You show that past research results using the Big Five, but are there other studies? If so, why not briefly present them, and make sure to justify the use of the Big Five rather than other personality theories.
Response: Thank you for your insightful comment. The majority of studies on personality and pro-environmental behavior have used the Big Five model, making it the dominant framework in this field. While alternative models exist, such as the HEXACO model, only a limited number of studies have applied it in the context of ecological behavior. We acknowledge this and have cited one study that employed HEXACO (Brick & Lewis, 2016) in our manuscript.
Our decision to use the Big Five model was based on its widespread application in prior research, allowing for direct comparisons with existing studies and contributing to a broader understanding of personality influences on pro-environmental behavior.
Comment: Materials and Method
Participants and procedure
You mention using G*Power to determine the required sample size. This is very useful, but it would be helpful to briefly explain why you chose a small effect size (f² = 0.02) ?
Response: Thank you for your insightful comment. Pro-environmental behavior is a multifaceted phenomenon influenced by numerous external and internal factors. The interplay of these variables often results in relatively small but significant effect sizes for individual predictors. By using a small effect size, we aimed to account for this complexity and avoid overestimating the impact of any single variable. This is our reason to adopt a conservative approach.
Comment: You mention that you planned to exclude some participants, what are the criterion of exclusion ?
Response: Thank you for your question. We recruited slightly more participants in case that some of the participants needed to be excluded due to incomplete or missing data, where participants fail to answer all required survey items, or in cases where participants do not meet the study’s demographic (e.g. being below 18 years old).
Comment: While you mention data collection through a university campaign and the use of Google Forms, it would be interesting to add more context regarding the duration of data collection ?
Response: Thank you for your question. The study was conducted from March 20, 2023, to September 19, 2023.
Comment: Have you considered the representativeness of the sample? If so, please developp it; if not, consider addressing it in the discussion.
Response: Thank you for your comment. Based on it, we included the following paragraph in the limitation part:
“Finally, the study's generalizability may be constrained by sample characteristics, as voluntary participation could have led to a self-selection bias favoring individuals already invested in environmental issues. Expanding research to include more diverse populations across different sociocultural and economic contexts would enhance the applicability of findings and provide a more comprehensive understanding of the psychological drivers of sustainability behaviors.”
Comment: Résults
« Previous studies have found links between Agreeableness and empathy, altruism, higher levels of selflessness, and greater concern for others [36, 37], which may explain the connection between this personality trait and pro-environmental behaviors. » Could you argument how is this connection possible ?
Response: Based on our comprehensive revision of the paper, this sentence has been omitted.
Reviewer 3 Report
Comments and Suggestions for Authors
In general, this manuscript is good and interesting for readers, but there are several things that can be improved to improve the quality of the manuscript:
1. on lines 64-68, it is better to explain several theories related to personality
2. the urgency of this research is not yet visible in the research objectives
3. based on the results of this study, the impact of changing personality has not been explained, does the person's pro-environmental behavior automatically change?
4. Are there other factors that may be the cause of changes in pro-environmental behavior that are still related to personality? should be explained in the discussion section
Comments on the Quality of English Languagenone
Author Response
Comment: on lines 64-68, it is better to explain several theories related to personality
Response: Thank you for your valuable suggestion. Based on the reviewers’ feedback, we have carefully revised our conceptual framework and methodology, incorporating additional variables such as all dimensions of environmental commitment (affective, verbal, and actual) and self-efficacy to enhance the depth of our analysis.
While personality traits remain an important component of our study, our revised approach places greater emphasis on environmental attitudes and self-efficacy as key predictors of pro-environmental behavior. Given this shift in focus, we have decided not to include an extensive discussion of multiple personality theories. Our rationale for this decision is that personality traits, although relevant, were found to have a comparatively weaker influence on pro-environmental behaviors than environmental attitudes.
Comment: the urgency of this research is not yet visible in the research objectives
Response: Thank you for your insightful comment. To address it, we have now made the urgency of this research more explicit in the research objectives as follows:
„Given the escalating environmental crisis and the need for behavioral changes at both individual and societal levels, understanding the psychological determinants of pro-environmental behavior is crucial. The aim of the current study is to explore pro-environmental behavior through a multidimensional approach, examining how personality traits, self-efficacy, and environmental attitudes (affective, actual, and verbal commitment) interact to shape various types of pro-ecological behaviors. Unlike previous research that has primarily examined general environmental engagement, this study differentiates between three distinct categories of pro-environmental behaviors: general behaviors, pro-social and volunteering actions, and household chemical usage. In doing so, it expands upon existing literature by considering how personality-driven tendencies interact with self-efficacy and environmental attitudes to predict various pro-environmental actions.“
Comment: based on the results of this study, the impact of changing personality has not been explained, does the person's pro-environmental behavior automatically change?
Response: Thank you for your comment. We appreciate the opportunity to refine our interpretation and strengthen our manuscript based on this valuable feedback.
Based on our revision and inclusion of new variables, our results indicate that environmental attitudes are the strongest predictors of general ecological behavior, whereas actual commitment, self-efficacy, and extraversion play a more significant role in prosocial engagement and volunteering within an ecological context. These findings challenge previous claims about the role of personality traits, suggesting that environmental attitudes and self-efficacy may be stronger predictors of sustainability-related behaviors than personality alone. We have revised the discussion section accordingly to clarify that behavioral change is not necessarily dependent on personality traits but can be effectively influenced by fostering environmental commitment.
Comment: Are there other factors that may be the cause of changes in pro-environmental behavior that are still related to personality? should be explained in the discussion section
Response: Thank you for your insightful comment. In revising the manuscript, we shifted our focus and incorporated additional variables. This adjustment allowed us to gain a more comprehensive understanding of the factors influencing pro-environmental behavior. As a result of these changes, our findings indicate that environmental attitudes have a significantly greater impact on pro-environmental behavior than personality traits.
Given this shift in focus and the new empirical insights, we decided not to elaborate extensively on personality traits in the discussion section. While personality remains relevant, our results suggest that environmental attitudes play a more direct role in shaping pro-environmental behavior. Thus, rather than expanding on multiple personality theories, we prioritized a more focused discussion that aligns with our revised findings.